Li *et al. Genome Biology*    (2020) 21:265

**METHOD**                                                **Open Access**

# The design and construction of reference pangenome graphs with minigraph

Heng Li[1,2]* , Xiaowen Feng[1,2] and Chong Chu[2]

*Correspondence:
hli@ds.dfci.harvard.edu
[1]Department of Data Sciences,
Dana-Farber Cancer Institute,
Boston 02215, MA, USA
[2]Department of Biomedical
Informatics, Harvard Medical
School, Boston 02215, MA, USA

**Abstract**

The recent advances in sequencing technologies enable the assembly of individual genomes to the quality of the reference genome. How to integrate multiple genomes from the same species and make the integrated representation accessible to biologists remains an open challenge. Here, we propose a graph-based data model and associated formats to represent multiple genomes while preserving the coordinate of the linear reference genome. We implement our ideas in the minigraph toolkit and demonstrate that we can efficiently construct a pangenome graph and compactly encode tens of thousands of structural variants missing from the current reference genome.

**Keywords:**  Bioinformatics, Genomics, Pangenome

## Background

The human reference genome is a fundamental resource for human genetics and biomedical research. The primary sequences of the reference genome GRCh38 [1] are a mosaic of haplotypes with each haplotype segment derived from a single human individual. They cannot represent the genetic diversity in human populations, and as a result, each individual may carry thousands of large germline variants absent from the reference genome [2]. Some of these variants are likely associated with phenotype [3] but are often missed or misinterpreted when we map sequence data to GRCh38, in particular with short reads [4]. This under-representation of genetic diversity may become a limiting factor in our understanding of genetic variations.

 Meanwhile, the advances in long-read sequencing technologies make it possible to assemble a human individual to a quality comparable to GRCh38 [1, 5]. There are already a dozen of high-quality human assemblies available in GenBank [6]. Properly integrating these genomes into a reference *pangenome*, which refers to a collection of genomes [7], would potentially address the issues with a single linear reference.

A straightforward way to represent a pangenome is to store unaligned genomes in a full-text index that compresses redundancies in sequences identical between individuals [8–10]. We may retrieve individual genomes from the index, inspect the k-mer spectrum and test the presence of k-mers using standard techniques. In principle, it is also possible to apply canonical read alignment algorithms to map sequences to the collection, but in practice, the redundant hits to multiple genomes will confuse downstream mapping-based analyses [11]. It is not clear how to resolve these multiple mappings.

The other class of methods encodes multiple genomes into a sequence graph, usually by collapsing identical or similar sequences between genomes onto a single representative sequence. The results in a *pangenome graph*. A pangenome graph is a powerful tool to identify core genome, the part of a genome or gene set that is shared across the majority of the strains or related species in a clade [12]. A common way to construct a basic pangenome graph is to generate a compacted de Bruijn graph (cDBG) [13–19] from a set of genomes. Basic cDBG does not keep sample information. Iqbal et al. [20] proposed colored cDBG with each color represents a sample or a population. Colored cDBG can be constructed efficiently [21, 22]. However, a colored cDBG discards the chromosomal coordinate and thus disallows the mapping of genomic features. It often includes connections absent from the input genomes and thus encodes sequences more than the input. A colored cDBG cannot serve as a *reference* pangenome graph, either. deBGA [23] addresses the issue by labeling each unitig with its possibly multiple locations in the input genome(s). Pufferfish [24] further reduces its space requirement. Nonetheless, given hundreds of human genomes, there will be many more vertices in the graph and most vertices are associated with hundreds of labels. Whether deBGA and pufferfish can scale to such datasets remains an open question. GBWT [25] provides another practical solution to storage and indexing, but no existing tools can practically construct a cDBG for many human genomes in the GBWT representation.

In addition to cDBG, we can derive a reference pangenome graph from a single linear multi-sequence alignment (MSA) [26, 27]. It has been used for HLA typing but is not applicable to whole chromosomes when they cannot be included in a single linear MSA. The third and possibly the most popular approach to reference graph generation is to call variants from other sources and then incorporate these variants, often in the VCF format [28], into the reference genome as alternative paths [29–33]. However, because VCF does not define coordinates on insertions, this approach cannot properly encode variations on long insertions and is therefore limited to simple variations. There are no satisfactory solutions to the construction of reference pangenome graphs.

In this article, we introduce the reference Graphical Fragment Assembly (rGFA) format to model reference pangenome graphs. We propose and demonstrate an incremental procedure to construct graphs under this model. The resulting graphs encode structural variations (SVs) of length 100bp or longer without haplotype information. Our implementation, minigraph [34] (https://github.com/lh3/minigraph), can construct a pangenome graph from twenty human assemblies in 3 h.

## Results

We will first describe a data model for reference pangenome graphs, which establishes the foundation of this article. We will then present a new sequence-to-graph mapper, minigraph, and show how this mapper incrementally constructs a pangenome graph. We

will demonstrate the utility of pangenome graphs with a human graph generated from twenty human haplotypes and a primate graph generated from four species.

### Modeling reference pangenome graphs

#### *Sequence graphs*

There are several equivalent ways to define a sequence graph. In this article, a *sequence graph G(V, E)* is a bidirected graph. Each vertex $v \in V$ is associated with a DNA sequence; each edge $e \in E$ has two directions, one for each endpoint, which leads to four types of edges: forward-forward, reverse-forward, forward-reverse, and reverse-reverse. The directions on an edge dictate how a sequence is spelled from a walk/path in the graph. Common assembly graphs, such as the overlap graph, string graph, and de Bruijn graph can all be formulated as sequence graphs.

The Graphical Fragment Assembly (GFA) format [35] describes sequence graphs. The core of GFA is defined by the following grammar:

```
<GFA>       <-(|<link>)+
   <-'S'<segId><segSeq>
<link>      <-'L'<segId>[+-]<segId>[+-]<cigar>
```

A line starting with letter "S" corresponds to a vertex and a line starting with "L" corresponds to a bidirected edge. In a de Bruijn graph, we often attach sequences to edges instead of vertices [36, 37]. To avoid the confusion, in this article, we also call a vertex as a *segment* and call an edge as a *link*, following the GFA terminology. Figure 1a shows an example GFA that encodes Fig. 1b.

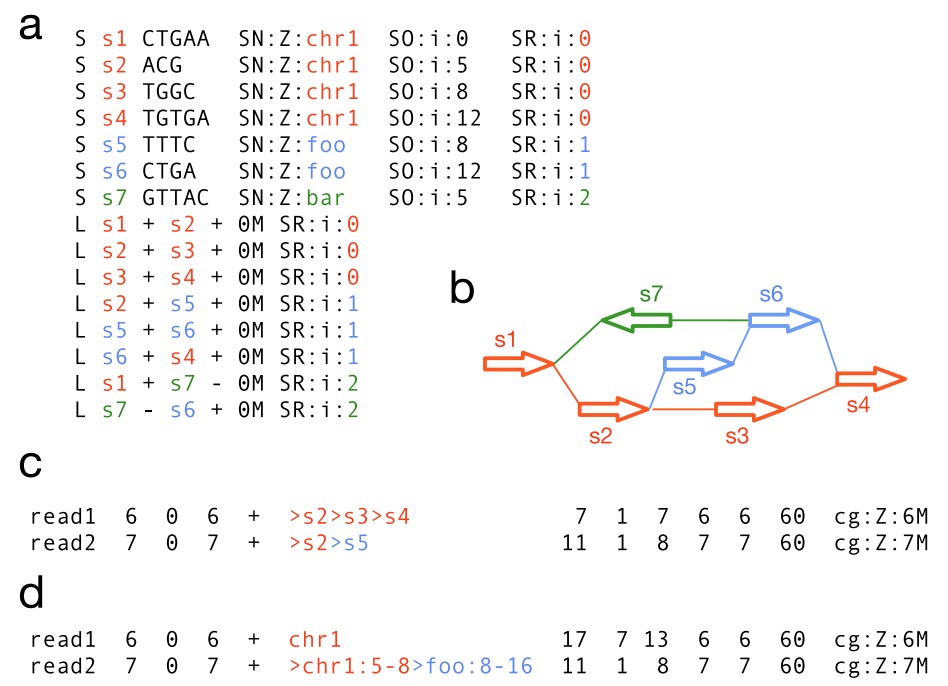

**Fig. 1** Example rGFA and GAF formats. **a** Example rGFA format. rGFA-specific tags include SN, name of the stable sequence from which the vertex is derived; SO, offset on the stable sequence; SR, rank: 0 if the vertex or edge is on the linear reference; >0 for non-reference. **b** Corresponding sequence graph. Each thick arrow represents an oriented DNA sequence. **c** Example GAF format, using the segment coordinate, for reads "GTGGCT" and "CGTTTCC" mapped to the graph. **d** Equivalent GAF format using the stable coordinate

A sequence graph in the GFA format natively defines a *segment coordinate* system where each base in the graph is uniquely indexed by a 2-tuple (segId, segOffset). For example, in Fig 1a, the base at position (s2, 2) is "G." A major problem with this coordinate is that it is decoupled from linear annotations and is sensitive to graph transformations. For example, if we split a segment into two connected segments, the set of sequences spelled from the graph remains the same, but the segment coordinates will be changed. Due to the instability of segment coordinate, a basic sequence graph is inadequate for a reference graph.

### Reference pangenome graphs

We propose the reference GFA (rGFA) format to encode reference pangenome graphs. rGFA is an extension to GFA with three additional tags that indicate the origin of a segment from linear genomes (Fig. 1a). This simple addition gives us a unique stable coordinate system as an extension to the linear reference coordinate (e.g., GRCh38). We can pinpoint a position such as "chr1 : 9" in the graph and map existing annotations onto the graph. We can also report a path or walk in the stable coordinate. For example, path "s1 → s2 → s3" unambiguously corresponds to "chr1:0-5 → chr1:5-8 → chr1:8-12" or simply "chr1:0-12" if we merge adjacent coordinate; similarly, "s1 → s2 → s5 → s6" corresponds to "chr1:0-8 → foo:8-16". We will formally describe the path format when introducing the GAF format in the next section.

In rGFA, each segment is associated with one origin. This apparently trivial requirement in fact imposes a strong restriction on the types of graphs rGFA can encode: it forbids the collapse of different regions from one sequence, which would often happen in a cDBG. We consider this restriction an advantage of rGFA because it requires the graph to have a "linear" flavor intuitively and simplifies the data structure to store the graph.

For simplicity, rGFA disallows overlaps between edges and forbids multiple edges (more than one edges between the same pair of vertices). These two restrictions help to avoid ambiguity and reduce the complexity in implementation. They are not strictly necessary in theory.

### The Graphical mApping Format (GAF)

As there are no text formats for sequence-to-graph alignment, we propose a new Graphical mApping Format (GAF) by extending the Pairwise mApping Format (PAF) [35]. GAF is TAB-delimited with each column defined in Table 1. Column 6 encodes a path on the graph. It follows the formal grammar below:

```
<path>       <- <stableId>|<orientIntv>+
<orientIntv> <- ('>'|'<')(<segId>|<stableIntv>)
<stableIntv> <- <stableId>':'<start>'-'<end>
```

In this grammar, `<segId>` is a segment identifier on an S-line in rGFA; `<stableId>` is a stable sequence name at the `SN` tag on the corresponding S-line. Column 6 can be either a path in the segment coordinate (Fig. 1c) or an equivalent path in the stable coordinate (Fig. 1d). We can merge adjacent stable coordinates if the two segments are originated from the same stable sequence and the end offset of the first segment is equal to the start offset of the second segment. For example, ">chr1:0-5>chr1:5-8" can be simplified to ">chr1:0-8". Furthermore, if a path in column 6 is derived from one

**Table 1** The Graphical mApping Format (GAF)

| Col | Type | Description |
|---|---|---|
| 1 | string | Query sequence name |
| 2 | int | Query sequence length |
| 3 | int | Query start coordinate (0-based; closed) |
| 4 | int | Query end coordinate (0-based; open) |
| 5 | char | Strand relative to col. 6 |
| 6 | string | Graph path matching regular expression |
|  |  | `/([><][^\s><]+(:\d+-\d+)?)+|([^\s><]+)/` |
| 7 | int | Path sequence length |
| 8 | int | Path start coordinate |
| 9 | int | Path end coordinate |
| 10 | int | Number of matching bases in the mapping |
| 11 | int | Number of bases, including gaps, in the mapping |
| 12 | int | Mapping quality (0–255 with 255 for missing) |

reference sequence, we recommend to replace it with the entire reference path on the forward orientation (e.g. see "read1" in Fig. 1d). With this convention, a GAF line is reduced to PAF for a sequence mapped to a reference sequence. Similar to PAF, GAF also allows optional tags in the SAM-like format. Base alignment is kept at the `cg` tag.

Minigraph produces GAF in both the segment and the stable coordinate. GraphAligner [38] produces GAF in the segment coordinate only, which can be converted to the stable coordinate.

### Sequence-to-graph mapping

Our incremental graph construction algorithm relies on genome-to-graph alignment (Fig. 2b). As existing sequence-to-graph aligners [38, 39] do not work with chromosome-long query sequences, we adapted minimap2 [40] for our purpose and implemented minigraph (Fig. 2a). Briefly, minigraph uses a minimap2-like algorithm to find local hits to segments in the graph, ignoring the graph topology. It then chains these local hits if they are connected on the graph, possibly through cycles. This gives the approximate mapping locations. Minigraph does not perform base-level alignment. This is because the graph we construct encodes SVs and rarely contains paths similar at the base level. The best mapping is often clear without base alignment.

To evaluate the accuracy of minigraph mapping, we simulated PacBio reads from GRCh38 with PBSIM [41] and mapped them to the graph we constructed in the next section. Table 2 compares the performance of minigraph and GraphAligner [38] v1.0.10 on 68,857 simulated reads mapped over 8 CPU threads. The N50 read length is 15kb. Nine thousand eight hundred sixty-two reads are mapped across two or more segments by GraphAligner. Note that both minigraph and GraphAligner ignore the stable coordinates during mapping. All segments, originated either from GRCh38 or from individual genomes, are treated equally. To this end, while we simulated reads from GRCh38, we are also evaluating how well mappers work with complex SVs present in any input samples.

On this dataset, minigraph is faster than GraphAligner and uses less memory, partly because minigraph does not perform base alignment. As is shown in Table 2, minigraph is more accurate than GraphAligner. This is counter-intuitive given that GraphAligner does base alignment. Close inspection reveals that most mismapped reads by minigraph

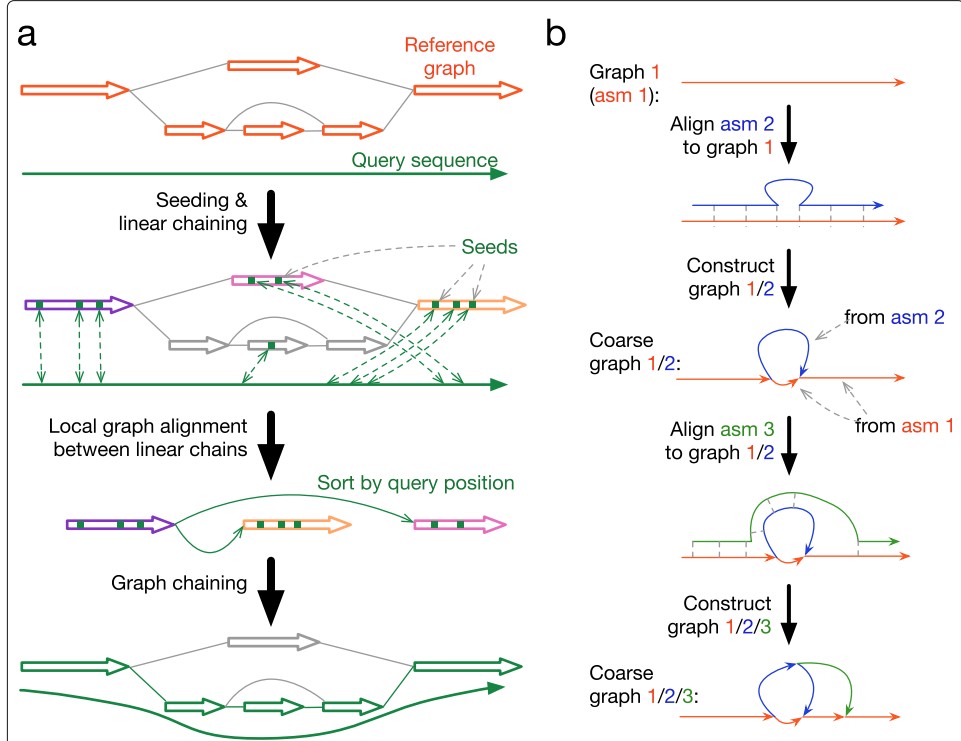

**Fig. 2** Minigraph algorithms. **a** Diagram of the minigraph mapping algorithm. Minigraph seeds alignments with minimizers, finds good enough linear chains, connects them in the graph and seeks the most weighted path as a graph chain. **b** Diagram of incremental graph construction. A graph is iteratively constructed by mapping each assembly to an existing graph and augmenting the graph with long poorly mapped sequences in the assembly

are mapped to the correct genomic loci but wrong graph paths. On the contrary, most mismapped reads by GraphAligner are mapped to wrong genomic loci. This suggests minigraph is better at finding approximate mapping locations but GraphAligner is better at disambiguating similar graph paths. Combining the strength of both could lead to a better graph mapper. We do plan to implement base-level alignment in minigraph in future.

We have also tried vg v1.21.0 [39]. It indexed the same graph in 14.7 wall-clock hours and mapped the simulated reads in 1.8 h over 8 threads, tens of times slower than minigraph and GraphAligner. However, no reads are mapped in the output. We have not been able to make vg work with our data.

**Table 2** Performance of sequence-to-graph mapping

|  | Minigraph | GraphAligner |
|---|---|---|
| Indexing time (wall-clock sec) | 100 | 589 |
| Mapping time (wall-clock sec) | 79 | 140 |
| Peak RAM (GB) | 19.5 | 27.2 |
| Percent unmapped reads | 0.5% | 0% |
| Percent wrong mappings | 1.7% | 4.6% |

### Generating pangenome graphs

Figure 2b shows how minigraph constructs a pangenome graph (see the "Methods" section for details). This procedure is similar to multiple sequence alignment via partial order graph [42] except that minigraph works with cyclic graphs and ignores small variants. Minigraph only considers SVs of 100 bp–100 kb in length and ignores SVs in alignments shorter than 100kb. For each input assembly, it filters out regions covered by two or more primary alignments longer than 20 kb in the assembly. This filter avoids paralogous regions in a sample and guarantees that graphs generated by minigraph can be modeled by rGFA.

As a sanity check, we compared minigraph to dipcall (https://github.com/lh3/dipcall) on calling SVs 100bp or longer from a synthetic diploid sample composed of CHM1 and CHM13 [4]. Given two SV callsets *A* and *B*, we say a call in *A* is *missed* in callset *B* if there are no calls in *B* within 1000bp from the call in *A*. With this criterion, 2.7% of 14,792 SVs called by dipcall are missed by minigraph; 6.0% of 14,932 minigraph SVs are missed by dipcall. We manually inspected tens of differences in IGV [43] and identified two causes. First, an INDEL longer than 100 bp called by one caller may be split into two shorter INDELs by the other caller. There are often more than one smaller SVs around a missed SV call. Second, dipcall skips regions involving high density of SNPs or involving both long insertions and long deletions, but minigraph connects these events and calls SVs in such regions. It tends to call more SVs. Overall, we believe minigraph and dipcall found similar sets of SVs.

### A human pangenome graph

Starting with GRCh38, we constructed a human pangenome graph from 20 human haplotypes or haplotype-collapsed assemblies (Table 3). It took minigraph 2.7 wall-clock hours

**Table 3** Assemblies used for graph construction

| Name | Species | Population | Accession/source |
| --- | --- | --- | --- |
| CHM1 | Human | N/A | GCA_001297185.1 |
| CHM13 | Human | N/A | GCA_000983455.1 |
| NA12878 | Human | European | [44], phased |
| NA24385 | Human | Jewish | [44], phased |
| PGP1 | Human | N/A | [44], phased |
| NA19240 | Human | African | GCA_001524155.4 |
| HG00514 | Human | East Asian | GCA_002180035.3 |
| HG01352 | Human | American | GCA_002209525.2 |
| NA19434 | Human | African | GCA_002872155.1 |
| HG02818 | Human | African | GCA_003574075.1 |
| HG03486 | Human | African | GCA_003086635.1 |
| HG03807 | Human | South Asian | GCA_003601015.1 |
| HG00733 | Human | American | GCA_002208065.1 |
| HG02059 | Human | East Asian | GCA_003070785.1 |
| HG00268 | Human | European | GCA_008065235.1 |
| HG04217 | Human | South Asian | GCA_007821485.1 |
| AK1 | Human | East Asian | GCA_001750385.1 |
| Clint | Chimpanzee | | GCA_002880755.3 |
| Susie | Gorilla | | GCA_900006655.3 |
| Kamilah | Gorilla | | GCA_008122165.1 |
| Susie | Orangutan | | GCA_002880775.3 |

over 24 CPU threads to generate this graph. The peak memory is 98.1GB. The resulting graph consists of 148,618 segments and 214,995 links. It contains 37,332 variations, where a *variation* denotes a minimal subgraph that has a single source and a single sink with both segments coming from GRCh38. A path through the bubble between the source and the sink represents an *allele*.

Variations in the human graph are enriched with Alus and VNTRs (Fig. 3a). While interspersed repeats are about evenly distributed along chromosomes except in the pseudoautosomal regions (Fig. 3e), VNTRs are enriched towards telomeres [6]. It is worth noting the density of minisatellites is also higher in subtelomeres. If we normalize the density of VNTRs in the pangenome graph by the density of minisatellites in GRCh38,

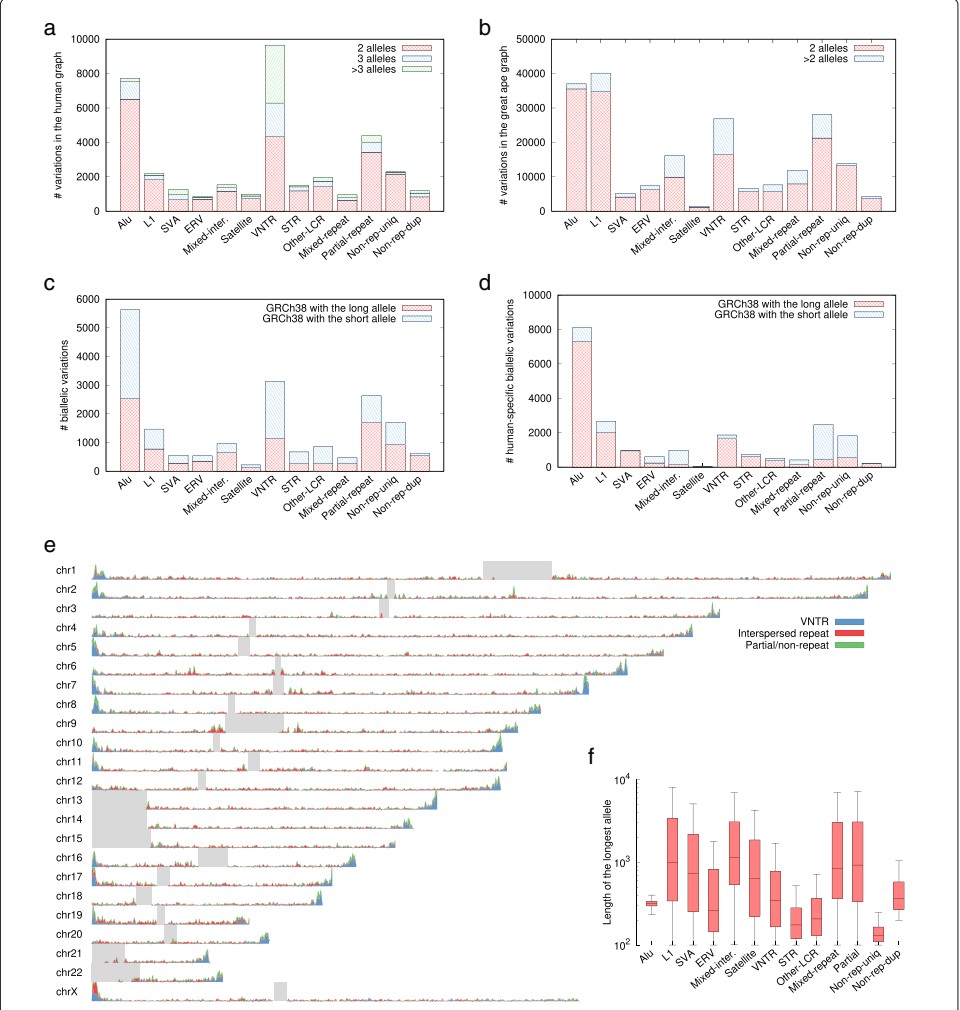

**Fig. 3** Characteristics of the human and the great ape graphs. **a** Human variations stratified by repeat class and by the number of alleles of each variation. The repeat annotation was obtained from the longest allele of each variation. VNTR: variable-number tandem repeat, a tandem repeat with the unit motif length ≥7bp. STR: short random repeat, a tandem repeat with the unit motif length ≤6bp. LCR: low-complexity regions. Mixed-inter.: a variation involving ≥2 types of interspersed repeats. **b** Great ape variations stratified by repeat class and by the number of alleles. **c** Human biallelic variations stratified by repeat class and by insertion to/deletion from GRCh38. Both alleles are required to be covered in all assemblies. **d** Human-specific biallelic variations stratified by repeat class and by insertion to/deletion from GRCh38. Red bars correspond to insertions to the human lineage. **e** Distribution of different types of human variations along chromosomes. **f** Boxplot of the longest allele length in each repeat class. Outliers are omitted for the clarity of the figure

the enrichment of VNTRs towards telomeres is still visible but becomes less prominent. At the same time, repeat-less variations are also enriched towards the ends of chromosomes (green areas in Fig. 3e), suggesting subtelomeres tend to harbor SVs anyway. We also identified 85 processed pseudogenes among these variations.

Another noticeable feature of VNTRs is that over half of VNTR variations are multi-allelic (Fig. 3a). Figure 4 shows a multi-allelic region composed of VNTRs. We can see many insertions of different lengths. The two different NA12878 assemblies also disagree with each other, which we often see around other VNTR loci in NA12878 as well. We have not inspected raw reads in this particular example, but we tend to believe the disagreement is caused by local misassemblies rather than somatic mutations. In addition, due to the multiallelic nature of such VNTRs, the two haplotypes in a human individual are often different. Assemblies mixing the two haplotypes (aka collapsed assemblies) may have more troubles in these regions. Multiallelic VNTRs are hard to assemble correctly.

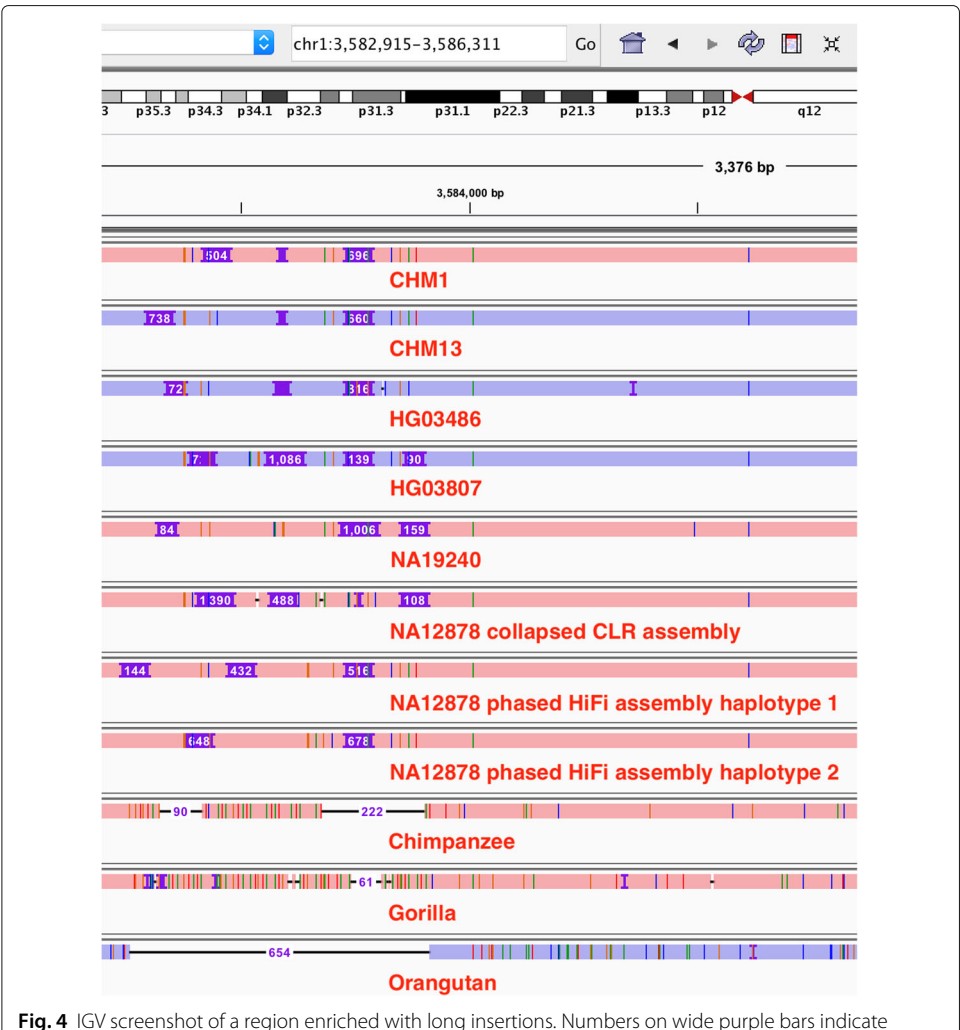

**Fig. 4** IGV screenshot of a region enriched with long insertions. Numbers on wide purple bars indicate insertion lengths. CLR: PacBio noisy continuous long reads. HiFi: PacBio high-fidelity reads

Multiallelic VNTRs are also hard to align and to call. In Fig. 4, the insertion positions are often different, which could be caused by a few mutations or sequencing errors. A naive alignment-based SV caller would call a dozen of low-frequency insertions in this region, which does not reflect these correlated events. Without base-level alignment, minigraph may have more troubles with obtaining the optimal alignment in these complex VNTR regions. Improved data quality, assembly algorithms and graph mapping algorithms are required to investigate VNTR regions in detail.

### A great ape pangenome graph

We also constructed a great ape pangenome graph from GRCh38, one chimpanzee, two gorillas, and one orangutan (Table 3). This graph contains 206,452 variations, over four times more than the human graph. About half of variations are originated from orangutan, the species most distant from human.

In the great ape graph, the L1-to-Alu ratio is close to 1:1, much higher than the ratio in the human graph (Fig. 3b vs a). This is perhaps correlated with the elevated L1 activity in great apes [45]. Of retrotransposon-related variations specific to the human lineage, the overwhelming majority are insertions (Fig. 3d), which is expected as transpositions lead to insertions only. Most human-specific Alu deletions are incomplete and involve ancient Alu subfamilies. They are likely genomic deletions that happen to hit Alus. In contrast, the majority of "partial-repeats" are deletions from the human lineage. Two thirds of autosomal insertions in this category are segmental duplications in GRCh38. In all, minigraph is an efficient tool to study closely related species.

### Blacklist regions from human pangenome graphs

The human pangenome graph effectively encodes SVs $\geq$100bp in 20 genomes. These large-scale variations could be a frequent source of technical artifacts in variant calling with short reads. To test this hypothesis, we compared short-read SNP calls with vs without regions around SVs in the pangenome graph.

We constructed a human pangenome graph excluding CHM1 and CHM13, the two samples used in the SynDip benchmark [4], and generated regions around variations (see the "Methods" section), which we call as *blacklist regions*, following the rationale in [46]. Blacklist regions is totaled 29.2 Mb in length, intersecting 0.7% of confident regions in SynDip [4]; 0.7% of truth SNPs are contained in blacklist regions—true SNPs are not enriched in blacklist regions.

We mapped short reads used in [4] with minimap2 and called variants with GATK v4.1.2 [47]. This callset contains 32,879 false positive SNPs, 21% of which fall in blacklist regions—false SNP calls are highly enriched in this < 1% region of human genome. This confirms a noticeable fraction of false SNP calls using short reads are resulted from misalignment involving SVs.

### Discussion

Based on the GFA assembly format [35], we proposed the rGFA format, which defines a data model for reference pangenome graphs at the same time. rGFA takes a linear reference genome as the backbone and maintains the conceptual "linearity" of input genomes.

rGFA is not the only pangenome graph model. Vg [39] encodes a stable sequence with a path through the sequence graph [48]. A segment in the graph may occur on multiple paths, or occur multiple times on one path if there are cycles in the graph. This way, vg allows different regions in one chromosome collapsed to one segment. We call such a graph as a collapsed graph. rGFA cannot encode a collapsed graph. The vg model is thus more general.

In our view, however, the reference pangenome graph should not be a collapsed graph. In a collapsed graph, the definition of orthology is not clear because multiple sequences from the same sample may go through the same segment. Without the concept of orthology, we cannot define variations, either. In addition, due to the one-to-many relationship between segments and the reference genome, it is intricate to derive the stable coordinate of a path in a collapsed graph. For example, suppose segment s1 corresponds to two regions chr1:100-200 and chr1:500-600. To convert a path s2 → s1 → s3 to the stable coordinate, we have to inspect adjacent segments to tell which s1 corresponds to; this becomes more challenging when s2 and s3 represent multiple regions in the reference genome. In contrast, rGFA inherently forbids a collapsed graph and avoids the potential issues above. This makes rGFA simpler than vg's path model and easier to work with.

To demonstrate practical applications of rGFA, we developed minigraph to incrementally generate pangenome graphs. It can generate a graph from 20 genomes in 3 h and can scale to hundreds of genomes in future. A limitation of minigraph is that it does not perform base alignment and may be confused by similar paths in the graph. Unfortunately, base-level sequence-to-graph alignment is not a fully solved problem. Partial-order graph alignment [42] and PaSGAL [49] only work with directed acyclic graphs (DAGs). Vg [39] uses a heuristic to unroll cycles but it is expotential in time in the worst case and for DAGs, its exact mode is tens of times slower than PaSGAL. Antipov et al. [50] proved that alignment against cyclic graphs can be done in polynomial time. GraphAligner [38] implements a fast quadratic algorithm for computing edit distance [51]. However, edit distance based alignment disallows long INDELs and is often inadequate for accurate variant calling. Jain et al. [52] recently proposed a quadratic algorithm for alignment with affine gap penalty but the authors focused on the theoretical analysis only. To the best of our knowledge, no tools can efficiently perform sequence-to-graph alignment under affine gap cost. We plan to learn from the existing algorithms and implement fast base alignment in minigraph in future. This may take significant effort.

Another limitation of minigraph is that it is unable to align sequences against a graph encoding all small variants. Such a graph will be composed of millions of short segments. Not indexing minimizers across segments, minigraph will fail to seed the initial linear chains. This limitation can only be resolved by completely changing the minigraph mapping algorithm. Nonetheless, small variants are easier to analyze with the standard methods. Incorporating these variants unnecessarily enlarges the graph, complicates implementations, increases the rate of false mappings [53], and reduces the performance of common tasks. There is also no known algorithm that can construct such a complex graph for hundreds of human genomes.

Minigraph does not keep track of the sample information as of now. To address this issue, we are considering to implement colored rGFA, similar to colored de Bruijn graphs [20]. In a colored rGFA, a color represents one sample. Each segment or link is associated with one or multiple colors, indicating the sources of the segment or the

link. Colors can be stored in an rGFA tag or in a separate segment/link-by-sample binary matrix [22]. The matrix representation may be more compact given a large number of samples.

We have shown minigraph can be a fast and powerful research tool to summarize SVs at the population scale and to study the evolution of closely related species. A more practical question is how a reference pangenome graph may influence routine data analysis. Here is our limited view.

We think a critical role a reference graph plays is that it extends the coordinate system of a linear reference genome. This allows us to annotate variations in highly diverse regions such as the human HLA and KIR regions. The existing pipelines largely ignore these variations because most of them cannot be encoded in the primary assembly of GRCh38.

The extended graph coordinate system further helps to consistently represent complex SVs. Given multiple samples, the current practice is to call SVs from individual samples and then merge them. Two subtly different SVs, especially long insertions, may be called at two distinct locations and treated as separate events. With the minigraph procedure, the two SVs are likely to be aligned together as long as they are similar to each other and are sufficiently different from the reference allele. To some extent, minigraph is performing multiple sequence alignment with partial order alignment [42]. This procedure is more robust to different representations of the same SV than naive merging. When we refer to a SNP, we often use its chromosomal coordinate such as "chr1:12345". We rarely do so for SVs because their positions are sensitive to alignment and SV callers. The more consistent SV representation implied by a pangenome graph will help to alleviate the issue and subsequently facilitate the genotyping of SVs [33, 54, 55].

While we believe a reference pangenome graph will make complex variations more accessible by geneticists and biologists, we suspect a great majority of biomedical researchers will still rely on a linear reference genome due to the conceptual simplicity of linear genomes and the mature tool chains developed in decades. Many analyses such as SNP calling in well behaved regions do not benefit much from a pangenome representation, either. Nonetheless, a pangenome reference still helps applications based on linear references. With a graph reference, we may blacklist regions enriched with SVs that lead to small variant calling errors. We may potentially generate "decoy" sequences that are missing from the primary assembly to attract falsely mapped reads away. We may perform read alignment against a graph, project the alignment to the linear coordinate and finish the rest of analyses in the linear space. We anticipate a pangenome reference to supplement the linear reference, not to replace it.

## Conclusions

Complex human sequence variations are like genomic dark matter: they are pervasive in our genomes but are often opaque to the assay with the existing tools. We envision a pangenome graph reference will become an effective means to the study of these complex variations. We proposed a data model (rGFA), designed formats (rGFA and GAF), and developed companion tools (minigraph and gfatools) to demonstrate the feasibility of our vision. Our work is still preliminary but it is likely to set a starting point to the development of the next-generation graph-based tools, which may ultimately help us to understand our genomes better.

## Methods

### The minigraph mapping algorithm

#### *Seeding and linear chaining*

Similar to minimap2, minigraph uses minimizers on segments as seeds. It also applies a similar chaining algorithm but with different scoring and with a new heuristic to speed up chaining over long distances. For the completeness of this article, we will describe part of the minimap2 chaining algorithm here.

**Minimap2-like chaining** Formally, an *anchor* is a 3-tuple $(x, y, w)$, representing a closed interval $[x - w + 1, x]$ on a segment in the reference graph matching an interval $[y - w + 1, y]$ on the query. Given a list of anchors sorted by $x$, let $f(i)$ be the maximal chaining score up to the $i$th anchor in the list. $f(i)$ can be computed by:

$$f(i) = \max \left\{ \max_{i > j \geq 1} \left\{ f(j) + \alpha(j, i) - \beta(j, i) \right\}, w_i \right\} \tag{1}$$

where $\alpha(j, i) = \min \left\{ \min \left\{ y_i - y_j, x_i - x_j \right\}, w_i \right\}$ is the number of matching bases between anchor $i$ and $j$. $\beta(j, i)$ is the gap penalty. Let $g_{ji} = \left| (y_i - y_j) - (x_i - x_j) \right|$ be the gap length and $d_{ji} = \min \left\{ y_i - y_j, x_i - x_j \right\}$ be the smaller distance between the two anchors. Minigraph uses the following gap cost:

$$\beta(j, i) = \begin{cases} \infty & (g_{ji} > G) \\ c_1 \cdot g_{ji} + c_2 \cdot d_{ji} + \log_2 g_{ji} & (0 < g_{ji} \leq G) \\ 0 & (g_{ji} = 0) \end{cases}$$

where $G = 100000$ in the graph construction mode, $c_1 = e^{-dw}$ and $c_2 = 0.05 \cdot e^{-dw}$. By default, $d = 0.01$ is the expected per-base sequence divergence and $w = 19$ is the minimizer length. In comparison, minimap2 applies $G = 5000$, $c_1 = 0.19$ and $c_2 = 0$. Minigraph allows much larger gaps between minimizers and more heavily penalizes gaps.

Solving Eq. 1 leads to an $O(n^2)$ algorithm where $n$ is the number of anchors. This algorithm is slow for large $n$. Minimap2 introduces heuristics to speed up the computation by approximating this equation. It works well for minimap2 that only allows small gaps and has base-level alignment as a fix to chaining errors. However, as minigraph intends to chain much longer gaps, the minimap2 algorithm occasionally misses the optimal alignment in long segmental duplications and produces false variations. Minigraph introduces a new heuristic to speed up chaining.

**Dynamic 1-dimension Range-Min-Query** Before we move onto the minigraph solution, we will first introduce Range-Min-Query (RMQ). Given a set of 2-tuples $\left\{ (y_i, s_i) \right\}$, $\text{RMQ}(a, b)$ returns the minimum $s_j$ among $\left\{ s_j : a \leq y_j \leq b \right\}$. We implemented 1-dimension RMQ with a modified AVL tree, a type of balanced binary search tree (Fig. 5). When performing $\text{RMQ}(a, b)$, we first find the smallest and the largest nodes within interval $[a, b]$ using the standard algorithm. In this example, the two nodes are (21,32) and (45,21), respectively. We then traverse the path between the two nodes to find the minimum. With a balanced tree structure, we do not need to descend into subtrees. The time complexity is $O(m \log m)$, where $m$ is the number of nodes in the tree. We can insert nodes to or delete nodes from the tree while maintaining the property of the tree. This achieves dynamic RMQ.

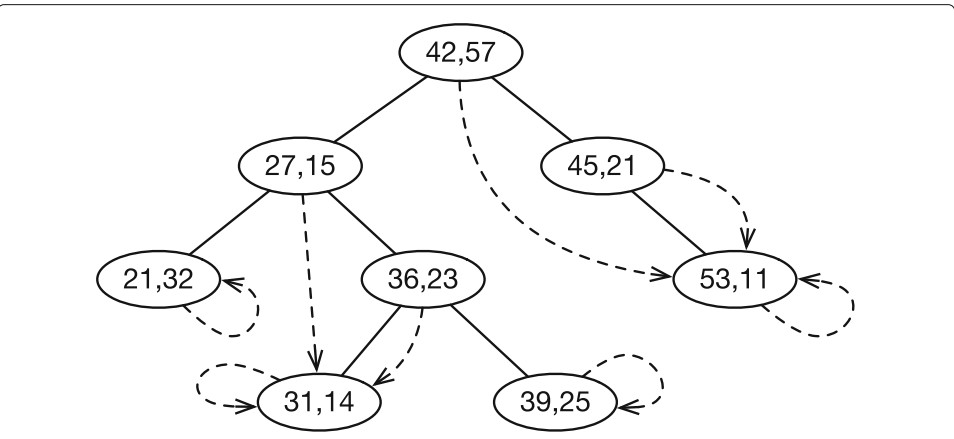

**Fig. 5** Implementing 1-dimension Range-Min-Query (RMQ). Given a set of 2-tuples, a binary search tree is built for the first values in the tuples. Each node $p$ in the tree is associated with a pointer. The pointer points to the node that is in the subtree descended from $p$ and has the minimal second value. In this example, RMQ(20, 50) = 14

**Chaining with a linear gap cost function**  A linear gap cost takes the form of $\beta'(j, i) = c_1 \left[ (y_i - y_j) + (x_i - x_j) \right]$. Given a list of anchors $(x_i, y_i, w_i)$ sorted by position $x_i$, let

$$f'(i) = \max_{\substack{i > j \geq 1 \\ x_i - G \leq x_j \leq x_i - w_i \\ y_i - G \leq y_j \leq y_i - w_i}} \left\{ f'(j) + w_j - \beta'(j, i) \right\} \tag{2}$$

We can find the optimal $f'(i)$ in $O(n \log n)$ time with RMQ [56, 57]. To see that, define

$$h'(j) = f'(j) + w_j + c_1 (y_j + x_j)$$

The following condition

$$f'(j) + w_j - \beta'(j, i) > f'(k) + w_k - \beta'(k, i)$$

is equivalent to $h'(j) > h'(k)$, independent of $i$. If we maintain $\mathrm{RMQ}_i$ as the binary tree that keeps $\left\{ (y_j, -h'(j)) : j < i, x_i - G \leq x_j \leq x_i - w_i \right\}$, we have

$$f'(i) = -\mathrm{RMQ}_i (y_i - G, y_i - w_i) - c_1 (x_i + y_i)$$

This solves Eq. 2 in $O(n \log n)$ time.

**Minigraph linear chaining**  While chaining with a linear gap cost function can be solved efficiently, we prefer more realistic cost function used in minimap2. In practical implementation, when we come to anchor $i$, we find the optimal predecessor $j_*$ under the desired gap cost $\beta(j, i)$ for anchors $\left\{ j : j < i, x_i - G' \leq x_j < x_i, y_i - G' \leq y_j < y_i \right\}$, where $G' < G$ is set to 10000 by default. Meanwhile, we use the RMQ-based algorithm to identify the anchor $j'_*$ optimal under the linear gap cost $\beta'(j, i)$. We choose $j'_*$ as the optimal predecessor if

$$f(j_*) + \alpha(j_*, i) - \beta(j_*, i) < f(j'_*) + \alpha(j'_*, i) - \beta(j'_*, i)$$

This may occasionally happen around long segmental duplications when the minimap2 heuristic misses the optimal solution. Effectively, minigraph does thorough search in a small window and approximate search in a large window using a faster but less sophisticated gap cost function.

### Graph chaining

Minigraph generates a set of linear chains $\{L_i\}$ with the procedure above that completely ignores the graph topology. It then applies another round of chaining taking the account of the topology.

We say linear chain $L_i$ *precedes* $L_j$, written as $L_i \prec L_j$, if (1) the ending coordinate of $L_i$ on the query sequence is smaller than the ending coordinate of $L_j$, and (2) there is a walk from $L_i$ to $L_j$ in the graph. If there are multiple walks from $L_i$ to $L_j$, minigraph enumerates the shortest 16 walks and chooses the walk with its length being the closest to the query distance between $L_i$ and $L_j$.

Given a list of linear chains sorted by their ending coordinates on the query sequence, let $g(i)$ be the optimal graph chaining score up to linear chain $L_i$. We can compute $g(i)$ with another dynamic programming:

$$g(i) = \max \left\{ \max_{L_j \prec L_i} \left\{ g(j) + \omega(L_j) - \beta(j,i) \right\}, \omega(L_i) \right\}$$

where $\beta(j,i)$ is the weight between $L_i$ and $L_j$. As minigraph does not perform base-level alignment, $\beta(j,i)$ is the same as the gap penalty function used for linear chaining. $\omega(L_i)$ is the optimal score of $L_i$ computed during linear chaining.

The procedure above has two limitations. First, when computing the weight between $L_i$ and $L_j$, minigraph largely ignores base sequences and only considers the distance between them on both the query and the graph. When there are multiple walks of similar lengths between $L_i$ and $L_j$, minigraph miss the graph chain that leads to the best base alignment. Although we added a heuristic by considering 17-mer matches between the query and the graph paths, we found this heuristic is not reliable in complex regions. Second, minigraph only enumerates the shortest 16 walks. In complex subgraphs, the optimal walk from $L_i$ to $L_j$ may not be among them. We plan to implement base alignment to address the limitations. We may use the current minigraph algorithm for easy cases and apply the more expensive base alignment when the current algorithm potentially fails.

The graph chaining algorithm results in one or multiple graph chains. A *graph chain* is a list of anchors $(s_i, x_i, y_i, w_i)$, where $[x_i - w_i + 1, x_i]$ on segment $s_i$ in the graph matches $[y_i - w_i + 1, y_i]$ on the query sequence. A graph chain satisfies the following conditions: if $i < j$, $y_i < y_j$; if $i < j$ and $s_i = s_j$, we have $x_i < x_j$; if $s_i \neq s_{i+1}$, the two segments are adjacent on the graph. It is an extension to linear chains.

### The minigraph graph generation algorithm

Using the minimap2 algorithm [40], minigraph identifies a set of *primary chains* that do not greatly overlap with each other on the query sequence. A region on the query is considered to be *orthogonal* to the reference if the region is contained in a primary chain longer than 100 kb and it is not intersecting other primary chains longer than 20 kb.

Minigraph scans primary chains in orthogonal regions and identifies subregions where the query subsequences significantly differs from the corresponding reference subsequences. To achieve that, minigraph computes a score $h_i$ for each adjacent pair of anchors $(s_i, x_i, y_i, w_i)$ and $(s_{i+1}, x_{i+1}, y_{i+1}, w_{i+1})$. Let $d_i^x$ be the distance between the two anchors on the graph and $d_i^y = y_{i+1} - y_i$ be the distance on the query sequence. $h_i$ is computed as

$$h_i = \begin{cases} -10 & \text{if } d_i^x = d_i^y \leq w_{i+1} \\ \eta \cdot \max\left\{ d_i^x, d_i^y \right\} & \text{otherwise} \end{cases} \tag{3}$$

where $\eta$ is the density of anchors averaged across all primary graph chains. Define $H\left(i,j\right) = \sum_{k=i}^{j} h_k$. A highly divergent region between the query and the graph will be associated with a large $H\left(i,j\right)$. Minigraph uses the Ruzzo-Tompa algorithm [58] to identify all maximal scoring intervals on list $(h_i)$, which correspond to divergent regions. In each identified divergent region, minigraph performs base alignment [40, 59] between the query and the graph sequences and retains a region if it involves an INDEL $\geq$ 100bp in length or a $\geq$ 100bp region with base-level identity below 80%. In Eq. 3, -10 is an insensitive parameter due to the downstream filtering. In the end, minigraph augments the existing graph with identified variations (Fig. 2b).

### Annotating variations

We applied RepeatMasker [60] v1.332 to classify interspersed repeats in the longest allele sequence of each variation. RepeatMasker is unable to annotate VNTRs with long motifs. It also often interprets VNTRs as impure STRs. Therefore, we did not use the RepeatMasker VNTR or STR annotations directly. Instead, we combined RepeatMasker and SDUST [61] results to collect low-complexity regions (LCRs). We identified pure tandem repeats composed of a motif occurring twice or more (implemented in https://github.com/lh3/etrf). An LCR is classified as VNTR if 70% of the LCR is VNTR; similarly, an LCR is classified as STR if 70% is STR; the rest are classified as "Other-LCR" in Fig. 3. The annotation script is available in the minigraph GitHub repository.

### Creating blacklist regions

For each variation in the graph, we extend its genomic interval on GRCh38 by 50bp from each end. We name this set of intervals as $I_0$. We align sequences inserted to GRCh38 against GRCh38 with "minimap2 -cxasm20 -r2k" and filter out alignments with mapping quality below 5. Let $I(a, b)$ be the set of GRCh38 intervals that are contained in alignments with identity between $a$ and $b$. The blacklist regions are computed by $I_0 \cup I(0, 0.99) \setminus I(0.998, 1)$, where "$\cup$" denotes the interval union operation and "$\setminus$" denotes interval subtraction.

## Supplementary information

---

**Additional file 1:** Review history.

---

### Acknowledgments

We are grateful to Benedict Paten and Erik Garrison for discussions on pangenome graphs. We thank minigraph users who have suggested features and helped to fix various issues.

### Peer review information

### Review history

The review history is available as Additional file 1.

### Authors' contributions

HL conceived the project, developed minigraph, and drafted the manuscript. XF did the pseudogene analysis. CC helped with RepeatMasker annotation. All authors helped to revise the manuscript. The authors read and approved the final manuscript.

### Funding

This work is supported by National Institutes of Health (NIH) grant U01HG010961 and R01HG010040.

**Availability of data and materials**

Minigraph is openly available at https://github.com/lh3/minigraph and zenodo [34]. This repository also includes the script to convert from the segment coordinate to the stable coordinate, to annotate variations, and to generate blacklist regions from the graph. The companion gfatools is available at https://github.com/lh3/gfatools. The human and the great ape graphs are hosted at http://ftp.dfci.harvard.edu/pub/hli/minigraph/. The NA12878, NA24385, and PGP1 phased assemblies were downloaded from http://ftp.dfci.harvard.edu/pub/hli/whdenovo/. Assemblies generated by McDonnell Genome Institute include GCA_001524155.4 for NA19240, GCA_002180035.3 for HG00514, GCA_002209525.2 for HG01352, GCA_002872155.1 for NA19434, GCA_003574075.1 for HG02818, GCA_003086635.1 for HG03486, GCA_003086635.1 for HG03486, GCA_003601015.1 for HG03807, GCA_002208065.1 for HG00733, GCA_003070785.1 for HG02059, GCA_008065235.1 for HG00268, and GCA_007821485.1 for HG04217. Other assemblies are available from GenBank under accession GCA_001297185.1 for CHM1 [2], GCA_000983455.1 for CHM13 [2], GCA_001750385.1 for AK1 [62], GCA_002880755.3 for chimpanzee Clint [63], GCA_900006655.3 for gorilla Susie [64], GCA_008122165.1 for gorilla Kamilah [63] and GCA_002880775.3 for orangutan Susie [63].

**Ethics approval and consent to participate**

Ethical approval was not needed for this study.

**Competing interests**

The authors declare that they have no competing interests.

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

## 