## [**Additional file 1** Review history. · Genome Biology]

Review History

First round of review

Reviewer 1

Were you able to assess all statistics in the manuscript, including the appropriateness of statistical tests used? No.

Were you able to directly test the methods? No.

Comments to author:

Li and coauthors describe a new algorithm and format to represent genome variation as a graph based on reference graph coordinates. This is a timely manuscript and represents another solid methodological contribution to genome analysis by the author.

The paper is clearly written. As someone without formal training in computer science I was able to understand the concepts and methodological outlines. I did not attempt to follow the methods section in detail, but I trust that it's sound and that other reviewers will have comments if there are issues with clarity or substance.

Pangenome graphs with reference coordinates as proposed here are a natural evolution to existing graph methods. They will facilitate more accurate read mappings than currently possible with a single reference sequence and thus allow much better analysis of structural variation.

The new algorithm, minigraph, is available and seems to have users (though I did not attempt to download and run it). It can assemble a graph from 20 whole human genomes in reasonable time (though more about that below). The authors applied the method to 20 human genomes and also constructed a pan-ape graph.

In the interest of review efficiency, I recently adopted a practice of only commenting on the most important aspects of a paper and will not go into smaller details. My main questions are:

1. Minigraph currently does not incorporate base-level alignment. Can this be done as a post-processing step? If not it seems to me that it would be essential to add it. Or I may be missing something, having only limited familiarity with how GraphAligner uses base-level mapping.
2. Minigraph currently ignores events <100bp. Would performance really be impacted so much by dropping that to 50 or perhaps even 20 bases? Or is this part of the design whereby the lack of base-level alignment precludes the discovery of smaller events?
3. Minigraph currently only considers alignments of >100kb (or maybe I misread that, but it "ignores" SVs within such alignments). Please clarify and explain the rationale, and possibly adjust those parameters (though I would expect them to be parameters that are user-settable).

Overall it seems to me that minigraph is a big step in the right direction but that it does not yet make graph genomes truly useful. I like the honesty in the Discussion and Conclusion and believe that this should be published quickly so that the process to get to truly useful graph genomes is accelerated.

Reviewer 2

Were you able to assess all statistics in the manuscript, including the appropriateness of statistical tests used? Yes. No additional tests required.

Were you able to directly test the methods? Yes.

Comments to author:

The primary concerns are:

Copy number variants, and variation within them, may be difficult to represent in the "linear" format of rGFA. This type of variation may end up being one of the more important classes of variants to represent in a PGG. It would be good to have a graphic showing how complex variation: large multi-allelic tandem repeats (e.g. lpa), are represented.

The sequence-to-graph mapping section is a little too sparse. It is unclear from the results what read length was simulated. This is relatively few reads to assess the ability for minigraph to map across SV (different edges). Next, reads should be simulated from at least one long read assembly -- some of the SV assessed in these genomes may be more complex than what is in the reference.

While the title of the manuscript is "design and construction of reference pangenome graphs", it would be beneficial to be the "design, construction, and use of reference pangenome graphs", in which some table is shown where some gain in knowledge, specificity, or any quantitative metric is achieved by using a PGG relative to the standard reference. Unfortunately the dilemma is that long reads map quite well across SV in the reference, and many breakpoints can be reconstructed from split reads (e.g. work by Sedlazeck & Schatz), but the regions that could potentially represent difficulty for mapping are almost certainly not in the assemblies used (e.g. no T2T assembly).

Two potential methods to show this are: (1) find reads simulated from the assemblies that map differently to the pgg versus the reference. This leaves the authors flexibility to define what 'differently' means. (2) Simulate complex genomic architectures that are difficult to map to in a genome, but straightforward in a PGG. HLA is a topical locus.

These can all be addressed in a resubmission. The manuscript states multiple times that base alignment would be good to perform to resolve similar paths. While this is not brought up as a requirement for submission, the lack of such functionality is a concern. Certain complex variants that are critical to new discoveries with long reads, for example the VNTR motif associated with schizophrenia (Song, Lowe, and Kingsley 2018) would potentially be missed without base alignment.

The section in the discussion regarding not encoding all small variants in the graph is important and so far not the focus of most current PGG papers.

Minor:

There are some vague statements including:

(Page 9) "This may occasionally happen around long segmental duplications when the minimap2 heuristic misses the optimal solution" -- this could be quantified for what is a long segmental duplication, and how many of these are in the genome.

(Page 9) "these two limitations have minor effects" -- this could be quantified for how many loci have > 16 paths between two points.

Editor

Q1.0.1: Both referees have raised concerns about base alignment functionality.

A1.0.1: Minigraph is primarily designed as a graph generator with read mapping as a byproduct. Minigraph does not need base-level alignment to generate graphs most of time. Base alignment may improve complex corner cases as we discussed in the manuscript, but the great majority of SVs are simple enough to resolve with the current minimizer-based algorithm. Several other kmer-based tools, such as MHAP, mash and mashmap, are widely used without base alignment, either.

As a matter of fact, sequence-to-graph base alignment is still unsolved. We have added a new paragraph in page 7 that reviews the current progress. In brief, no existing algorithms can do fast affine-gap alignment we need. We expect a series of new papers coming in this direction. We are also thinking of a new sequence-to-graph alignment algorithm ourselves, but it will take significant effort to experiment it. We see that as a separate project.

Furthermore, our manuscript already represents a significant advance without base alignment. We are the first to build a human pangenome graph. We introduced a graph model and a format that may be used by others. As review 1 correctly pointed out, we don't anticipate this manuscript to solve all problems in pangenome analyses.

Reviewer #1

Li and coauthors describe a new algorithm and format to represent genome variation as a graph based on reference graph coordinates. This is a timely manuscript and represents another solid methodological contribution to genome analysis by the author.

The paper is clearly written. As someone without formal training in computer science I was able to understand the concepts and methodological outlines. I did not attempt to follow the methods section in detail, but I trust that it's sound and that other reviewers will have comments if there are issues with clarity or substance.

Pangenome graphs with reference coordinates as proposed here are a natural evolution to existing graph methods. They will facilitate more accurate read mappings than currently possible with a single reference sequence and thus allow much better analysis of structural variation.

The new algorithm, minigraph, is available and seems to have users (though I did not attempt to download and run it). It can assemble a graph from 20 whole human genomes in reasonable time (though more about that below). The authors applied the method to 20 human genomes and also constructed a pan-ape graph.

In the interest of review efficiency, I recently adopted a practice of only commenting on the most important aspects of a paper and will not go into smaller details. My main questions are:

Q1.1.1: Minigraph currently does not incorporate base-level alignment. Can this be done as a post-processing step? If not it seems to me that it would be essential to add it. Or I may be missing something, having only limited familiarity with how GraphAligner uses base-level mapping.

A1.1.1: GraphAligner is designed to be a read mapper. Minigraph is designed to be a graph generator from genome assemblies. They have different purposes. Base alignment is not essential to minigraph. Base alignment can be done as a post-processing step, but this wouldn't improve mapping accuracy. We plan to more properly implement base alignment in future. See also **A1.0.1**.

Q1.1.2: Minigraph currently ignores events <100bp. Would performance really be impacted so much by dropping that to 50 or perhaps even 20 bases? Or is this part of the design whereby the lack of base-level alignment precludes the discovery of smaller events?

A1.1.2: Minigraph is unable to seed across multiple segments in the graph and it requires at least two seeds on one segment to initiate linear chaining. As a result, minigraph will not be able to seed a 20bp event. Minigraph may not be able to seed a 50bp event if there are a couple of variants on it. Not seeding small events may affect mapping accuracy. We once built a graph with 50bp as the threshold. The graph looked actually fine. Nonetheless, we felt smaller events are less important biologically but make the graph more complex. We opted for the 100bp threshold in the end.

Q1.1.3: Minigraph currently only considers alignments of >100kb (or maybe I misread that, but it "ignores" SVs within such alignments). Please clarify and explain the rationale, and possibly adjust those parameters (though I would expect them to be parameters that are user-settable).

A1.1.3: Minigraph can map 150bp short reads (though the accuracy won't be as good as vg). The 100kb threshold is only used in the graph generation mode. Note that the contig N50 of input assemblies are all longer than 10Mb. Contigs shorter than 100kb often have low quality.

Q1.1.4: Overall it seems to me that minigraph is a big step in the right direction but that it does not yet make graph genomes truly useful. I like the honesty in the Discussion and Conclusion and believe that this should be published quickly so that the process to get to truly useful graph genomes is accelerated.

A1.1.4: We thank the reviewer for identifying our strength.

Reviewer #2

Li, Feng, and Chu present a method for representation, construction, and storing pan-genome graphs.

The primary concerns are:

Q1.2.1: Copy number variants, and variation within them, may be difficult to represent in the "linear" format of rGFA. This type of variation may end up being one of the more important classes of variants to represent in a PGG. It would be good to have a graphic showing how complex variation: large multi-allelic tandem repeats (e.g. lpa), are represented.

A1.2.1: All assemblies used in the manuscript failed to assemble the LPA region. To study LPA, we used our own hifiasm assembler to produce haplotype-resolved assemblies of CHM13, NA24385, HG00733, NA12878 and NA19240. Multiple assemblies go through the LPA region. We rebuilt a graph from these nine haplotypes. Fig. A1.2.1 shows the minigraph subgraph in this region. We can see multiple insertions, deletions and one large inversion (the rightmost angled blue bar). We are not confident in the graph construction in this region due to these overlapping complex events, but we believe our rGFA model can work with LPA.

Please note that hifiasm and these assemblies are unpublished. Fig. A1.2.1 is just a temporary result. Therefore, we are not showing this example in the manuscript. The Human Pangenome Project will produce haplotype-resolved assembly of 40 samples using our hifiasm assembler. Users will see a better LPA subgraph by then. We anyway thank the reviewer for the LPA example. This is an interesting case.

Fig. A1.2.1: Subgraph around the LPA locus. Thick red bars indicate segments from GRCh38. Blue bars come from a 94.3kb region on CHM13. The thin orange line shows how the path goes through the blue bars on CHM13. The thick gray bar comes from the paternal haplotype of NA24385/HG002.

Q1.2.2: The sequence-to-graph mapping section is a little too sparse. It is unclear from the results what read length was simulated.

A1.2.2: We now clarify in the text that the N50 read length is 15kb.

Q1.2.3: This is relatively few reads to assess the ability for minigraph to map across SV (different edges).

A1.2.3: Out of the 68,857 simulated reads, GraphAligner mapped 9,862 reads to two or more segments/nodes. There are enough reads mapped across segments.

Q1.2.4: Next, reads should be simulated from at least one long read assembly -- some of the SV assessed in these genomes may be more complex than what is in the reference.

A1.2.4: For long read or contig mapping, minigraph ignores the stable coordinates in rGFA. All segments, either coming from GRCh38 or from individual genomes, are treated equally by minigraph. To this end, while we simulated reads from GRCh38, we have already evaluated how minigraph aligns through complex SVs that are present in any samples. We have clarified this point in the revised manuscript on page 4.

Q1.2.5: While the title of the manuscript is "design and construction of reference pangenome graphs", it would be beneficial to be the "design, construction, and use of reference pangenome graphs", in which some table is shown where some gain in knowledge, specificity, or any quantitative metric is achieved by using a PGG relative to the standard reference. Unfortunately the dilemma is that long reads map quite well across SV in the reference, and many breakpoints can be reconstructed from split reads (e.g.

work by Sedlazeck & Schatz), but the regions that could potentially represent difficulty for mapping are almost certainly not in the assemblies used (e.g. no T2T assembly).

Two potential methods to show this are: (1) find reads simulated from the assemblies that map differently to the pgg versus the reference. This leaves the authors flexibility to define what 'differently' means. (2) Simulate complex genomic architectures that are difficult to map to in a genome, but straightforward in a PGG. HLA is a topical locus.

A1.2.5: We thank the reviewer for the suggestion. In the manuscript, we have shown that the graph we built helped to reduce false SNP calls (section "Blacklist regions from human pangenome graphs"). In addition, we have discussed the possibility of SV genotyping in page 8. The existing SV genotypers are incompatible with our graph representation but we expect future tools to take advantage of our graph.

Q1.2.6: These can all be addressed in a resubmission. The manuscript states multiple times that base alignment would be good to perform to resolve similar paths. While this is not brought up as a requirement for submission, the lack of such functionality is a concern. Certain complex variants that are critical to new discoveries with long reads, for example the VNTR motif associated with schizophrenia (Song, Lowe, and Kingsley 2018) would potentially be missed without base alignment.

A1.2.6: Most long variants, including VNTR, differ in length. Minigraph can tell which path to follow. Minigraph also implements a k-mer based heuristic to check sequences on different paths. For two paths of similar length, minigraph prefers the path of high identity. In addition, even with base alignment, VNTR is still hard to align (Fig. 4 in the manuscript). See also **A1.0.1**.

The section in the discussion regarding not encoding all small variants in the graph is important and so far not the focus of most current PGG papers.

Minor:

There are some vague statements including:

Q1.2.7: (Page 9) "This may occasionally happen around long segmental duplications when the minimap2 heuristic misses the optimal solution" -- this could be quantified for what is a long segmental duplication, and how many of these are in the genome.

A1.2.7: How often the minimap2 heuristic fails depends on the frequency of SVs involving long segmental duplications, which is largely unknown. It is not determined by the number of long segmental duplications in the reference genome.

Q1.2.8: (Page 9) "these two limitations have minor effects" -- this could be quantified for how many loci have > 16 paths between two points.

A1.2.8: During mapping, minigraph can seed long segments in the graph (see **A1.1.2**) and uses a k-mer based heuristic to check sequence similarity (see **A1.2.6**). It is hard to quantify how often the minigraph heuristic fails given practical sequences. We thank the reviewer for suggestions anyway.

Second round of review

Reviewer 2

Most of my concerns are addressed. There is a sticking point: the response to the critique: "these two limitations have minor effects" -- this could be quantified for how many loci have > 16 paths between two points.

was:

A1.2.8: During mapping, minigraph can seed long segments in the graph (see A1.1.2) and uses a k-mer based heuristic to check sequence similarity (see A1.2.6). It is hard to quantify how often the minigraph heuristic fails given practical sequences. We thank the reviewer for suggestions anyway

The response is lackluster. Unless the statement "have minor effects" is quantified, it is difficult to interpret what that means, and the statement does not add value/information to the manuscript.

Q2.0.1: Many thanks for submitting the revised version of your manuscript to Genome Biology. Please accept my apologies for the delay in getting back to you about it. It has now been seen again by Referee 2, whose comments are listed below. Once the outstanding concerns from Referee 2 and some remaining editorial concerns are satisfactorily addressed, we would, in principle, be delighted to publish your manuscript.

We ask that you address Referee 2's comment and modify the statement regarding the limitations of the approach.

A2.0.1: We have revised the manuscript to address the reviewer #2's remaining concern. Please see answer **A2.2.1** below for details.

Q2.0.2: As you may be aware, Genome Biology operates very strict open access, open source and open data policy. Therefore we ask that the method is deposited in a public software repository, such as GitHub, with the access link included. We ask that the source code is released under an open source license compliant with OSI (<http://opensource.org/licenses>) and that the license is clearly stated both on the repository site and in the manuscript. A version of source code used in the manuscript should also be deposited in a DOI-assigning repository, such as zenodo, with the DOI referenced. All this information should be included in the Availability of Data and Materials section of the manuscript.

Please provide a reference for your source code in zenodo, and for your github repository, and include these in the reference list, as with other references. This should be in the following format: Last, First. [all authors] Title. Github. hyperlink/doi (2019).

A2.0.2: Our code is already open sourced and publicly available at github. We have created a Zenodo DOI for the version of minigraph we were using for the manuscript. The software is cited as reference [34] in the revised manuscript.

Q2.0.3: Please also provide the accession numbers and include citations for the assemblies used in the study in the Availability of data and materials section.

A2.0.3: The accession numbers were provided in Table 3 in the original manuscript. We have added these accessions to the Data Availability section along with their citations.

Q2.0.4: Additionally, we have made some minor changes to the title and the abstract of the manuscript, to make them better adhere to Genome Biology's house style. Please find the edited versions pasted at the foot of this email. We ask that you include these versions in any subsequent submissions of your manuscript. Regarding the title, we mainly would like 'minigraph' mentioned in it in some form, so we are open to changes if you would prefer an alternative.

A2.0.4: We have changed the title and the abstract to your version. We thank the editor for the suggestions.

Q2.0.5: Please ensure that the manuscript contains the following sections: Ethical Approval section. If ethical approval was not needed for the study, please state in the manuscript that these are not applicable.

A2.0.5: We have added an Ethical Approval section, claiming that this approval was not needed for this study.

Q2.0.6: Author list not matching EM

- Please note the corresponding author Heng Li; hli@ds.dfci.harvard.edu listed in the manuscript is the primary corresponding author who will receive the e-proofs from production once it has been copyedited and typesetted. Only this author can view the eproofs and submit their corrections online. Please express if there are any other corresponding authors that should receive a copy of the proofs.

A2.0.6: Heng Li is the only corresponding author of this manuscript.

Q2.0.7: We are now able to include authors' twitter handles in the manuscript. If you or any of your co-authors would like to do this, please include the information as a separate affiliation for the author.

A2.0.7: We choose not to include twitter handles in this manuscript.

Review #2:

Q2.2.1: Most of my concerns are addressed. There is a sticking point: the response to the critique: these two limitations have minor effects" -- this could be quantified for how many loci have > 16 paths between two points.

was:

A1.2.8: During mapping, minigraph can seed long segments in the graph (see A1.1.2) and uses a k-mer based heuristic to check sequence similarity (see A1.2.6). It is hard to quantify how often the minigraph heuristic fails given practical sequences. We thank the reviewer for suggestions anyway

The response is lackluster. Unless the statement "have minor effects" is quantified, it is difficult to interpret what that means, and the statement does not add value/information to the manuscript.

A2.2.1: At the early development stage, we have experimented to use more paths. We observed little change to the resulting graph. We are confident that our heuristic doesn't impact the overall characteristics of the graph. Nonetheless, we haven't done this experiment using the latest version of minigraph. We agree with the reviewer that claiming "minor effects" without quantitative data is weak. We thus removed the sentence mentioning "have minor effects" in revision and just say: "we plan to implement base alignment to address the limitations".